# Temporal misalignment in scene perception: Divergent representations of locomotive action affordances in human brain responses and DNNs

**Clemens G. Bartnik (c.g.bartnik@uva.nl)**
Institute for Informatics, University of Amsterdam, Amsterdam, The Netherlands

**Evelyne I.C. Fraats**
Psychology Research Institute, University of Amsterdam, Amsterdam, The Netherlands

**Iris I.A. Groen (i.i.a.groen@uva.nl)**
Institute for Informatics, University of Amsterdam, Amsterdam, The Netherlands
Psychology Research Institute, University of Amsterdam, Amsterdam, The Netherlands

## Abstract

**The human visual system processes scenes with remarkable speed, enabling the extraction of essential information to navigate our surroundings in a single glance. To elucidate how the brain transforms visual inputs into neural representations of navigationally relevant information, we collected electroencephalography (EEG) responses to diverse indoor and outdoor scenes along with behavioral annotations of locomotive action affordances (e.g., walking, cycling), object annotations, and low-level image features to model distinct types of scene information. Using representational similarity analysis, we examined the neural representation of locomotive action affordances over time, their co-localization within scene-selective cortex, and their computational alignment with deep neural networks (DNNs). Our results show that locomotive action affordance representations emerge within 200 ms of visual processing, showing unique contributions to EEG responses at temporally distinct time-points from objects and low-level properties. Spatiotemporal fusion with functional magnetic resonance imaging (fMRI) recordings in scene-selective brain regions reveals that both the parahippocampal and occipital place region (but not the medial place region) contribute to locomotive action affordance representations, with a distinct temporal hierarchy between them. While DNNs align well with early EEG responses, they primarily capture low-level features and show limited alignment with affordance processing. These findings reveal a temporally distinct neural representation of action affordances and highlight a limitation of current DNNs in modeling affordance perception.**

**Keywords:** Navigational affordances; Scene perception; Electroencephalography; Representational Similarity Analysis; Spatiotemporal fusion; Deep neural networks

## Introduction

Humans effortlessly navigate dynamic environments, where changes can occur in milliseconds—such as cycling through dense morning traffic. This highlights the brain's remarkable speed in processing visual scenes (Fei-Fei et al., 2007; Potter, 1975; Thorpe et al., 1996), allowing it to capture relevant information to guide navigational actions in a glance (Greene

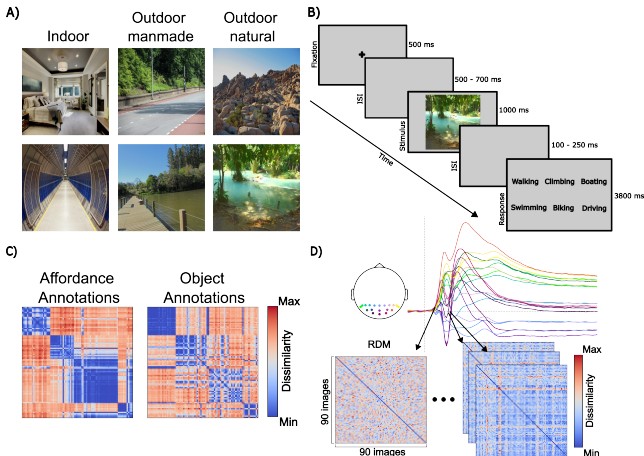

Figure 1: Overview: (A) Example scenes. (B) Experimental design. (C) RDMs from action affordance (left) and object annotations (right), with red indicating high and blue low dissimilarity. (D) Single-image evoked responses from 19 occipital electrodes, with RDMs computed from ERP response amplitudes every ∼8 ms (-100 to 1000 ms relative to image onset).

& Oliva, 2009). The process of identifying potential relevant interactions with the environment is broadly known as *affordance perception* (Gibson, 1977). Although the presence of scene-selective cortical regions is well established (Epstein & Baker, 2019; Dilks et al., 2022; Bartnik & Groen, 2023), and growing evidence highlights their sensitivity to navigational affordances (Bonner & Epstein, 2017; Dwivedi, Cichy, & Roig, 2021; Bartnik et al., 2025), most prior research has relied on fMRI measurements. As a result, the temporal dynamics of navigational affordance perception and the cascade of underlying neural computations remain poorly understood.

Studies using time-resolved brain measurements during scene perception consistently indicate that scene feature representations emerge as early as 100 ms after image onset, seemingly following a temporal hierarchy. Low-level features, such as global scene properties (e.g., clutter level, scene size, and overall spatial layout), are extracted between 90–150 ms (Groen et al., 2013; Ramkumar et al., 2016; Cichy et al., 2017;

Hansen et al., 2018). Open vs. closed scene discrimination (Lowe et al., 2018) and manmade vs. natural scene categorization (Groen et al., 2013; Harel et al., 2016) also emerge around that time. (Greene & Hansen, 2020) showed that information about objects contained in the scenes is processed around 175–225 ms, while correlations with behavioral tasks, especially affordance-based scene sorting, appear at later stages (Greene & Hansen, 2020). Recent studies show that spatial structure is processed between 90–125 ms, semantic content between 140–175 ms (Mononen et al., 2025), while navigational affordances may be processed even later, around 300 ms post-onset (Dwivedi et al., 2024).

This suggests that affordance perception might build upon previously extracted scene features. However, isolating individual contributions is challenging due to feature interdependencies (Groen et al., 2017; Malcolm et al., 2016), and it has been suggested that scene processing may not follow a straightforward temporal cascade from low- to high-level scene properties (Groen et al., 2017; Ramkumar et al., 2016; Greene & Hansen, 2020). Indeed, one study found that neural representations of local features diagnostic of navigational affordances emerge as early as 134 ms after image onset (Harel et al., 2022), and perceived affordances strongly shape scene categorization (Greene et al., 2014), inherently affecting how the environment itself is perceived (Djebbara et al., 2019). This view of affordances as a 'visual primitive' is supported by evidence that affordance perception is largely task-independent (Bonner & Epstein, 2017; Bartnik & Groen, 2023), though prior knowledge may still influence scene perception (Djebbara et al., 2019; Naveilhan et al., 2024).

When navigational affordance representations emerge in visual processing may depend on the way they are operationalized. Most existing studies focus on indoor environments, defining affordances as possible pathways through space (Bonner & Epstein, 2017; Dwivedi et al., 2024; Harel et al., 2022). We recently proposed an alternative approach considering locomotive action affordances, focusing on specific *types* of actions required to navigate environments (Bartnik et al., 2025). In this work, fMRI recordings revealed representation of such affordances in the scene-selective regions Parahippocampal Place Area (PPA) and Occipital Place Area (OPA). These representations were not explained away by low-level global scene features or object labels and were poorly captured by features learned by modern DNNs (Bartnik et al., 2025). Given these findings, we hypothesized that temporally-resolved neural measurements will also reveal unique locomotive action affordance representations that do not overlap with other visual features or DNN features. Such measures furthermore allow determining the temporal onset of locomotive affordance representations, which could help elucidate the temporal dynamics of affordance perception.

To test this hypothesis, we collected human electroencephalography (EEG) responses to our diverse set of images also used in Bartnik et al. (2025), and related these responses to behavioral annotations, fMRI data, and DNN feature activa-

tions. Our results confirm that locomotive action affordances form a unique representational space not only in fMRI but also in EEG signals; moreover, we find that they are processed within 200 ms but later than low-level global scene features or object representations, are temporally aligned with scene-selective regions, with a processing hierarchy showing earlier responses in OPA compared to PPA. While DNNs exhibit strong representational alignment with EEG signals, their representations align more closely with low-level features, suggesting they may be insufficient for accurately modeling the neural processing of locomotive action affordances.

## Methods and Materials

### Participants

Twenty healthy volunteers with normal or corrected vision (6 males; age 18–27, $M$ = 21.45, $SD$ = 2.43) participated in the study, which was approved by the Ethical Committee of the institution. They provided informed consent and received research credits or monetary compensation. Two participants were excluded due to incomplete data.

### Stimuli

The stimuli consisted of 90 high-resolution (1024×1024 pixels) color photographs, sourced from a copyright-free image database (Flickr) and previously used in our neuroimaging study (Bartnik et al., 2025). Each image was captured from a human-scale, eye-level perspective and depicted an everyday scene without humans or prominent central objects, and belonged to one of three environmental categories: indoor, outdoor-natural, or outdoor-man-made (see **Fig. 1A**).

### Experimental design and Procedure

The experiment consisted of three blocks, each corresponding to a different task (action, object, or fixation) which were presented in a counterbalanced order. In each block, participants viewed the same 90 images, each repeated six times. Once per image per block, a response screen appeared after the presentation of the image, prompting participants to press one of six keyboard buttons corresponding to task-specific labels: action affordances (walking, biking, driving, swimming, boating, climbing), contained objects (building, plant, water, furniture, road, stones), or the color of the fixation cross (blue, red, orange, purple, yellow, cyan). Image order and response option order were randomized separately for each participant. The response screen appeared randomly after one of the six repetitions, resulting in six task-specific repetitions and 18 total repetitions per image across tasks. Before each block, participants completed a training session with 15 practice images not included in the main experiment.

Stimuli (1024x1024 px) were presented using PsychoPy (v3.2.4) on a 2560×1440 px screen (59.6 × 33.6 cm) in a controlled lighting environment. Participants sat ∼70 cm from the screen, with images spanning 20° of their visual field. A fixation cross was displayed at the start (8 s) and end (12 s) of the experiment. Each trial followed this sequence: fixation cross (500 ms), blank grey screen (500–750 ms), stimulus image

with a randomly colored fixation cross (1000 ms), and another blank grey screen (100–250 ms). For response trials, a response screen appeared (max 3800 ms), followed by a jittered inter-stimulus interval (ISI) of 2000–4000 ms. Non-response trials proceeded directly to the ISI. Breaks were allowed after every 90 images and between task blocks.

## EEG acquisition and pre-processing

EEG data was recorded using a Biosemi 64-channel Active Two EEG system (www.biosemi.com) with a 10-20 layout at a sampling frequency of 2048.0 Hz. To capture more vision-related activity, two frontal electrodes (F5 & F6) were repositioned posteriorly to the left and right of Iz (renamed I1 & I2). Eye movements were monitored with electro-oculograms (EOGs). Preprocessing for the purpose of computing event-related responses (ERPs) was done in Python, MNE (Gramfort et al., 2013) and included the following steps. High-pass filter at 0.1 Hz (6 dB/octave); a low-pass filter at 30 Hz (6 dB/octave) (one-pass, zero-phase, non-causal bandpass filter, hamming window, 0.0194 passband ripple, 53 dB stopband attenuation, -6 dB cut-off frequency); two notch filters (zero-phase) at 50 Hz and 60Hz; epoch segmentation from -100 to 1000 ms from stimulus onset, downsampled to 128 Hz for computational efficiency; baseline correction between -100 and 0 ms; ocular correction using the EOG electrodes (Gratton et al., 1983); conversion to Current Source Density responses (Perrin et al., 1987). Artifacts were rejected using maximal allowed amplitudes of -75 and +75 µV in the Oz channel. This led to a rejection of 2.04% of the epochs with (mean = 5.5%, SD = 0.45% images per participant and task).

Trials with the same image were averaged across all repetitions to obtain non-task-specific ERPs for each image and each subject. For task-specific ERPs, only the image repetitions within each block corresponding to the same task were averaged. Following previous studies that demonstrated that navigational affordance processing happens in the visual cortex (Bonner & Epstein, 2017; Harel et al., 2022; Dwivedi et al., 2024), we selected 19 posterior and occipital channels for further data analysis (P1, P3, P5, P7, P9, PO7, PO3, O1, O2 Oz, POz, Pz, P2, P4, P6, P8, P10, PO4, PO8).

## RDM construction

Trial-averaged ERP responses were used to construct representational dissimilarity matrices (RDMs) for each subject using pairwise Pearson correlation distances between ERP amplitudes across 19 electrodes. This was done in a time-resolved manner at 7.86 ms intervals from -100 to 1000 ms relative to image onset (**Fig. 1D**). RDMs were computed using the Python version of the RSA toolbox (van den Bosch et al., 2025). We smoothed the data using sliding window (39.29 ms) averaging while maintaining dimensional consistency through padding. The effect of smoothing window size is illustrated in **Fig. S1B**. Since various distance metrics exist (e.g., A. Walther et al. (2016)), we compared multiple metrics (**Fig. S1C**).

## Behavioral Annotations

RDMs were calculated per participant by computing Euclidean distances between scenes based on the proportion of selected labels—six locomotive actions (e.g., walking, biking, driving) for affordances, and six object categories (e.g., buildings, roads, vegetation). These RDMs were then averaged across participants to obtain group-level behavioral RDMs.

## fMRI data

fMRI-based RDMs were constructed from multivoxel activation patterns of 20 separate participants viewing the same 90 stimuli (see Bartnik et al. (2025) for details). Activations were extracted from three scene-selective ROIs: PPA, OPA, and MPA, identified using separate category localizer scans. Voxel patterns were averaged across runs and used to compute subject-specific RDMs for each ROI using pairwise Pearson correlation distances between images.

## DNN feature activations

We extracted layer activations from several pre-trained deep neural network (DNN) models using the Net2Brain Python package https://github.com/cvai-roig-lab/Net2Brain (Bersch et al., 2022). We adopted the same model and layer selection as described in Bartnik et al. (2025). We used the layers pre-selected by Net2Brain, and created RDMs by standardizing the features within each layer (mean removal and scaling to unit variance) and computing pairwise distances between the flattened feature activations in each layer using Pearson correlation distance.

## Representational similarity analysis

To compare the EEG responses with behavioral annotations, fMRI responses and DNN layer feature representations, we used representational similarity analysis (RSA) (Kriegeskorte, 2008). Model RDMs were compared to the ERP RDMs for each individual participant and at each time point by computing Spearman's $\rho$ as proposed by the RSA toolbox (Kriegeskorte, 2008; Nili et al., 2014) (lower triangle, excluding the diagonal; Ritchie et al. (2017)). At each time point, one-sample t-tests were conducted to determine whether the average correlation across participants differed significantly from zero. P-values were corrected for multiple comparisons using fdr correction at alpha level 0.05. To compare peak correlation time points, paired t-tests were used, while independent t-tests were applied for DNN comparisons. All comparisons were fdr-corrected across all time-points in the ERP epoch.

## Variance Partitioning and Partial correlations

We used variance partitioning via regression to identify the unique and shared explained variance of each source. Therefore, we calculated the difference between the variance explained when all model RDMs were included as independent variables and the variance explained when all except the current model RDM were used as independent variables in a multiple linear regression model aimed at predicting the variance in ERP RDMs. Similar results were obtained when controlling

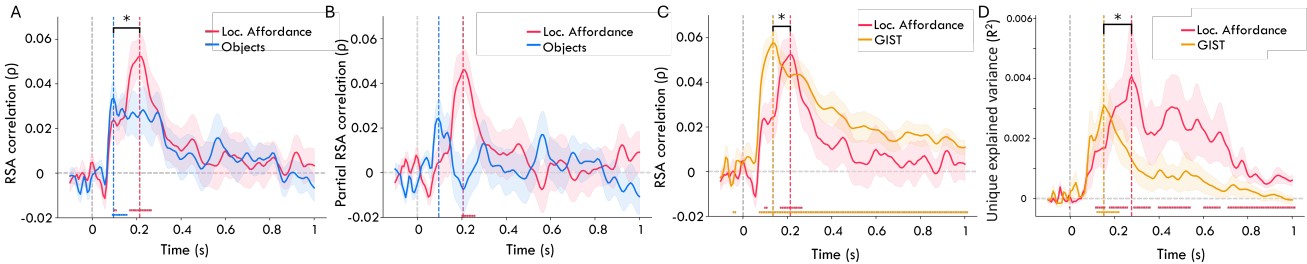

Figure 2: (A) Time-resolved, across-subject averaged RDM correlations between behavioral annotations and ERPs; shaded areas indicate standard error of the mean (SEM) across participants. Significant time points (fdr-corrected) are marked with dots, and vertical dashed lines highlight peak correlation times. (B) Partial correlations between action affordance and ERP RDMs, controlling for object representations, and vice versa. (C) Time-resolved correlations between ERP and GIST RDMs vs. locomotive action affordance RDMs. (D) Variance partitioning of action affordance and GIST representations in the EEG signal.

for the number of regressors using a shuffling approach that preserved model dimensionality by replacing predictor RDMs with condition-shuffled versions (see **Fig. S2**). This analysis was performed for each participant and time point separately. We then averaged the unique explained variance across participants and calculated the standard error of the mean (SEM). We also computed partial Spearman correlations by setting various model RDMs as covariates.

For the analyses quantifying the impact of controlling for fMRI RDMs or DNN RDMs on correlations between the affordance, object and GIST RDMs and EEG responses, we computed difference curves, whereby we subtracted the partial Spearman correlation time courses from the original correlation time courses. We then computed the Area under the Curve (AUC) for these difference curves using the trapezoidal rule for the full ERP epoch (-100ms to 1000ms). AUC values were compared using paired-sample t-tests for fMRI ROI comparisons, and Mann-Whitney *U* Test for DNN comparisons.

## Results

We recorded EEG as human participants viewed 90 real-world scene images spanning indoor, outdoor man-made, and outdoor natural environments (**Fig. 1A**). Participants completed three task blocks: 1) categorizing scenes on six locomotive action affordances (**Fig. 1B**), 2) identifying objects, and 3) reporting fixation cross color (control task). To quantify perceived locomotive action affordance and object representations, we computed pairwise dissimilarities between the obtained behavioral annotations, generating representational dissimilarity matrices (RDMs) for both affordance and object representational spaces (**Fig. 1C**). We computed EEG RDMs based on pairwise dissimilarities between event-related potential (ERP) amplitudes across posterior electrodes recorded for each scene, at each time point between -100ms to +1000ms relative to stimulus onset (**Fig. 1D**). Similar to Bartnik et al. (2025), task-based analyses revealed no robust effects of task instructions on neural responses (see **Fig. S3**), so all results reported are based on task-averaged ERPs.

## Unique locomotive action affordance representations emerge around 200 ms

To investigate when locomotive action affordance representations emerge during visual processing, we correlated the task-averaged ERP RDMs per time point with the behavioral RDMs for both locomotive action affordance and objects. The resulting correlation time courses (**Fig. 2A**) show that object representations emerged relatively early, around 100ms after image onset, while action affordance representations emerged later, peaking shortly after 210ms. A paired-sample t-test comparing peak correlation times across participants confirmed a significant delay in the processing of locomotive action affordances relative to objects (t(17) = 2.40, *p* = 0.028). To test whether the observed effects were driven solely by a walking versus non-walking distinction, we also compared a binary RDM separating these two classes, and found a significant peak around 250 ms (see **Fig. S4**), but the overall weaker and negative correlations suggest this contrast alone does not account for the affordance representations.

While these results provide initial evidence of a temporally distinct representation of locomotive action affordances in EEG responses, the two behavioral RDMs also show considerable inter-correlation (ρ = 0.60) (similar to our previous study (Bartnik et al., 2025); see **Fig. S1A** for a direct comparison). To better understand the individual contributions of each type of representation, we computed partial correlations, removing the shared contribution of the object representational RDM from the action affordance RDM, and vice versa. Controlling for object annotations reduces the correlation with action affordance at early time-points but maintains the distinct peak around 200 ms (**Fig. 2B**). This peak, along with subsequent time points, remained significant (all [*t* >3.59, all *p* <0.002; fdr-corrected], indicating processing of unique locomotive action affordance-related information. In contrast, controlling for locomotive action affordances rendered the object correlation peak at 100 ms insignificant, and also reduced correlations with object representations at later time points, revealing the opposite pattern as observed for action affordances.

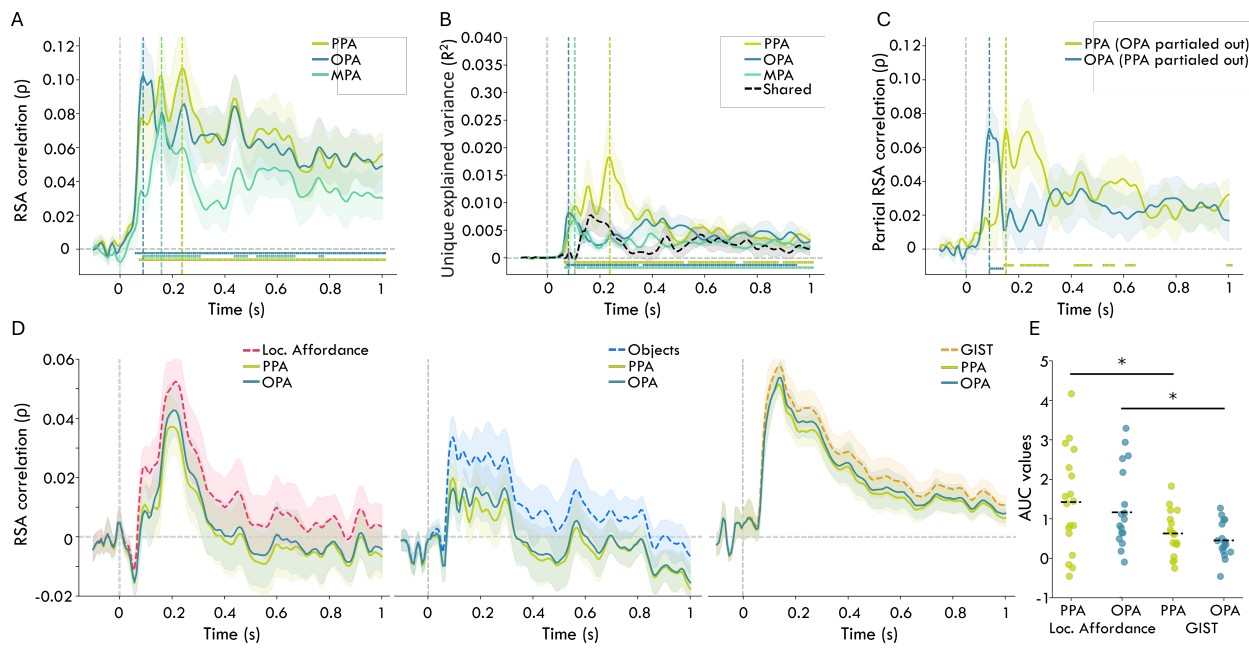

Figure 3: (A) Time-resolved, across-subject averaged RSA correlations between scene-selective ROI and ERP RDMs, with shaded areas indicating SEM. Significant time points (fdr-corrected) are marked with dots, and vertical dashed lines indicate peak correlation times. (B) Variance partitioning of RDM correlations (C) Partial Spearman correlations between PPA and ERP RDMs, controlling for OPA representations, and vice versa. (D) Left: Original correlation between locomotive action affordances and ERP RDMs (dashed lines), and correlations after controlling for PPA and OPA (solid lines). Middle and right: same but for objects and GIST correlations, respectively. (E) AUC values of difference curves from (D), quantifying the effect of subtracting partial correlations from original feature space correlations.

To account for image features beyond object content, we also compared to the GIST model (Oliva & Torralba, 2001), a well-established model of low-level features that captures global scene structure using Gabor–like filters across spatial frequencies, orientations, and locations, and has been shown to correlate well with early EEG signals (Greene & Hansen, 2020; Groen et al., 2017). As shown in **Fig. 2C**, correlations with GIST model RDMs emerge significantly earlier (peak at 135ms) than affordance representations (one-sample t-test: t(17) = 2.39, $p$ = 0.022). Partial correlation analysis produced nearly identical curves (**Fig. S5**), reflecting the fact that these representational spaces are not highly correlated ($\rho$ = 0.08) to another.

Interestingly, while **Fig. 2C** suggests a stronger correlation with GIST throughout the ERP time course, a variance partitioning analysis provided a clear distinction of each feature space's unique contribution to the EEG signal (**Fig. 2D**), with GIST features and affordances both accounting for unique variance between 150 and 200 ms, but affordances continuing to explain unique variance at later time points. These results show that locomotive action affordances are represented in EEG responses and are processed later than objects or GIST features.

**Scene-selective regions match the time course of locomotive action affordance representations**

In our prior fMRI study Bartnik et al. (2025) we demonstrated that scene-selective regions OPA and PPA, but not the Medial Place Area (MPA), automatically extract representations of locomotive action affordances. Our recordings of EEG responses to the same image set enables spatiotemporal fusion (Cichy et al., 2019) to assess each brain region's temporal correspondence with the ERP responses.

For this, we constructed average RDMs by calculating pairwise correlation distances of multi-voxel activity patterns from PPA, OPA, and MPA (see Bartnik et al. (2025) for further details on the fMRI analysis). Next, we correlated these three fMRI ROI RDMs with the ERP RDMs. **Fig. 3A** shows that all three scene-selective regions exhibit significant correlations with ERP responses around 90ms after stimulus onset (one-sample t-tests against 0, fdr-corrected for multiple comparisons). OPA peaks first at approximately 90ms, followed by MPA at 150ms. PPA shows an initial peak at 90ms but exhibits additional peaks at 150ms and 210ms.

Because all three regions are known to be involved in scene perception, they likely exhibit a substantial covariance. To identify their unique correspondence to the EEG signal, we again used variance partitioning to determine how much

unique variance each scene-selective region's RDM explains in the ERPs. **Fig. 3B** shows that indeed all three regions contain overlapping representations, indicated by the shared explained variance peaking around 150 ms. Nevertheless, each region also explains unique variance, starting from around 100 ms, with OPA and MPA contributing the most unique variance around this time. In contrast, PPA accounts for more unique variance at later time points, particularly after 200 ms.

While analyzing unique variance helps disentangle the individual contributions of each scene-selective region, the significant time windows remain highly overlapping. Indeed, despite the apparent differences in the time points of the highest average unique explained variance, a paired-samples t-test comparing the peak times for OPA and PPA showed no significant difference ($t(17) = -0.83$, $p = 0.417$). However, a more focused comparison using partial correlations between OPA and PPA reveals distinct temporal dynamics (**Fig. 3C**): OPA exhibits a unique partial correlation peak around 90ms post-image onset (all $t > 3.62$, all $p < 0.002$; fdr-corrected) indicating its distinct contribution at this early stage. Conversely, PPA's early correlations disappear when controlling for OPA, with significant peaks emerging only later, between 180–250ms post-onset (all $t > 2.71$, all $p < 0.015$; fdr-corrected). These non-overlapping time windows suggest that OPA processes information earlier, while PPA becomes engaged at later stages.

Thus far, we demonstrated temporal alignment between scene-selective ROI representations and EEG responses. But to what extent does this temporal alignment reflect the processing of action affordances, versus other image properties? To answer this question, we examined how correlations of affordance representations with ERPs change when controlling for OPA and PPA representations. In **Fig. 3D**, the first panel shows that partialing out either OPA or PPA RDMs reduces locomotive action affordance correlations with ERPs, suggesting both regions contribute to the affordance representations. The second panel shows a similar pattern for objects, with correlations reducing when partialing out both regions. In contrast, the GIST model correlations with ERPs remain largely unchanged when controlling for OPA and PPA (final panel), indicating minimal correspondence of these regions with the representation of GIST features in the ERPs.

To quantify these effects, we computed difference curves by subtracting the correlation time courses obtained by partialing out the fMRI ROIs (**Fig. 3D**) from the original correlation time courses, and then calculated the area under the curve (AUC) (see Methods), reflecting the extent of shared variance with the different feature representations. The resulting AUC values (**Fig. 3E**) were significantly larger when partialing scene-selective regions out of affordance representations in ERPs, compared to the GIST representations (PPA ($t(17) = 4.656$, $p < 0.001$) or OPA ($t(17) = 4.310$, $p < 0.001$). This shows that both regions indeed contribute to the ERP variance related to affordance processing, but not to GIST-related variance.

Overall, these results show that scene-selective regions account for variance in ERP signals, suggesting a processing hierarchy where OPA processes information earlier than PPA. Additionally, these regions appear to primarily share variance related to high-level features such as locomotive action affordances and object representations, rather than low-level features captured by the GIST model.

## Pre-trained DNNs correlate well with ERPs but show alignment with low-level features rather than locomotive action affordances

DNNs serve as effective models of human visual processing (Kietzmann et al., 2019), but their ability to encode features relevant to navigational affordances remains unclear. While prior research suggests they capture affordance-related information in the form of pathways (Bonner & Epstein, 2018), our earlier fMRI study (Bartnik et al., 2025) found stronger alignment with object representations. Here, we assess how well DNNs capture navigational affordances by examining their alignment with emerging visual representations in ERP responses over time.

**Fig. 4A** confirms the general notion that DNNs are good models capturing representations in visual cortex, as all tested DNN models, when averaging feature activations across layers, display substantial RSA correlations with the ERPs starting at ~100 ms (see **Fig. S6** for layer-specific results). Further, we observe that Transformer architectures (ViT) and models trained on image-text pairs (CLIP) show somewhat higher correlations than classic CNNs trained on object recognition, indicating higher representational alignment with brain activity.

To what extent are these DNNs able to capture the neural representation of locomotive action affordances in the EEG signal? To examine this, we first compared the peak correlation times for activations extracted from each individual DNN layer with the average peak time across subjects, for each of the three representational spaces we tested so far (affordances, objects and GIST; **Fig. 4B**). DNN model groups show no consistent ordering of peak timing, with all models showing a peak correlation in between the maxima for the GIST model and the object features; none of the models shows a clear temporal co-localization with the peak for affordances.

This initial comparison provides a first indication that DNN features primarily align with earlier visual processing stages that reflect low-level feature or object processing, rather than affordance processing. However, this peak misalignment does not provide a complete picture; RSA correlations with DNN features extend throughout the ERP epoch, and could thus still capture affordance representations at later stages. To compare representations across the entire ERP time course, we performed an analogous partial correlation analysis to the fMRI comparisons above. We reasoned that a significant reduction in correlation between ERPs and affordances when partialing out DNN activations would indicate representational overlap. We indeed observe a slight reduction when partialing out DNN activations from the locomotive action affordance correlations with the ERPs (**Fig. 4C**, left panel). ViTs and

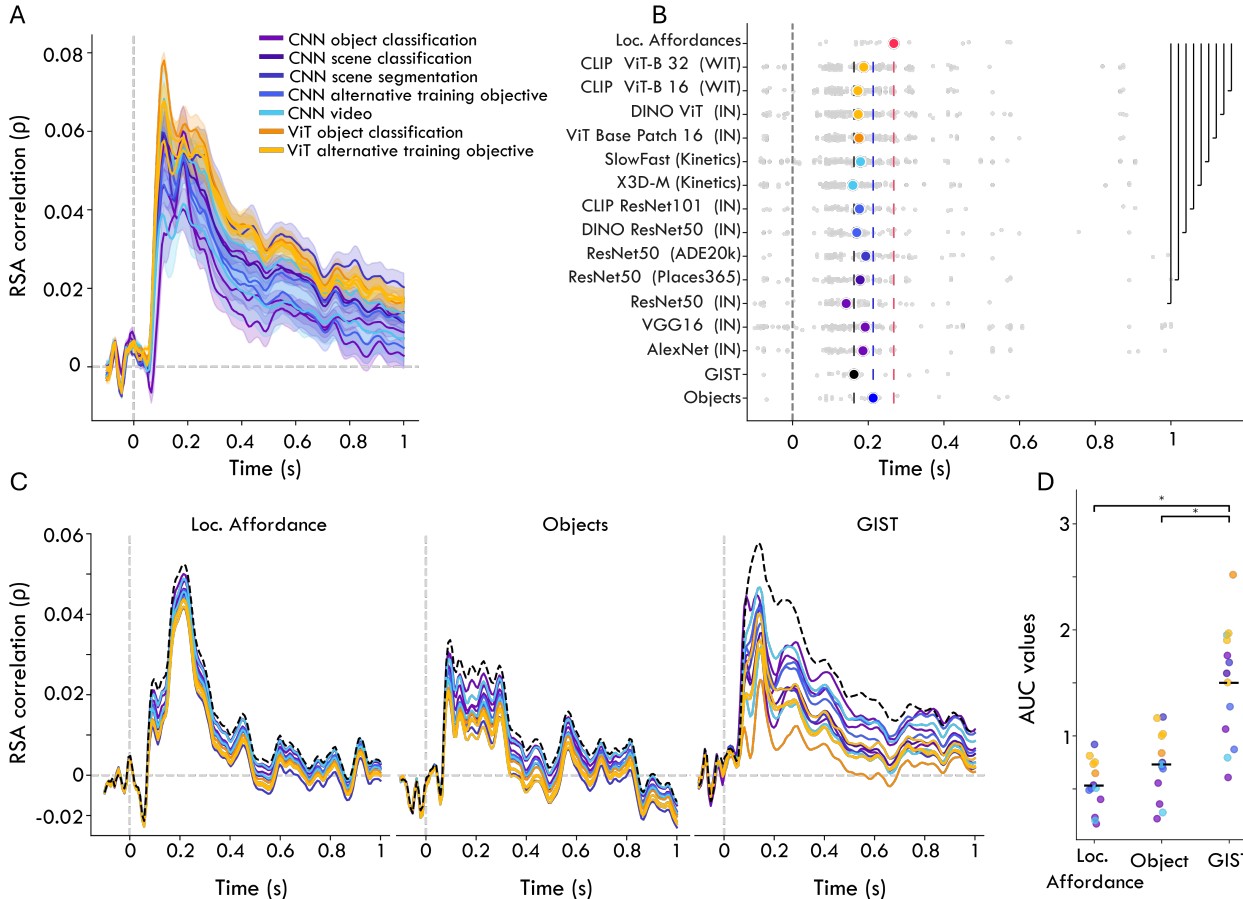

Figure 4: (A) Time-resolved RSA correlations between ERP RDMs and DNN feature RDMs (averaged across subjects and layers); shaded areas indicate SEM across layers. Significant time points (fdr-corrected) are marked with dots, and vertical dashed lines indicate peak correlation times. (B) Peak correlation time points for DNNs, with gray dots representing individual layer correlations and colored dots showing the average. Affordance, object and GIST model peak averages are marked with colored vertical markers, and significant differences from action affordance peak correlations (fdr-corrected) are highlighted with black vertical lines. (C) Time-resolved partial correlations between ERP RDMs and affordance (left), object (middle) and GIST (right) RDMs, after controlling for DNN features from each model shown in (A). The dashed black lines indicate the original RSA correlations. (D) AUC values of difference curves from (C), subtracting partial correlations from original correlations.

CLIP models induce the largest reductions, consistent with their overall stronger alignment with ERPs. Object correlations with ERPs show a similar, slightly stronger reduction, especially in early time points (**Fig. 4C**, middle panel). In comparison, GIST model correlations exhibit a much larger reduction when controlling for DNN activations, indicating more overlap of DNN representations with GIST features in ERPs than with affordances or objects (**Fig. 4C**, right panel).

To quantify these reductions, we again calculated the AUC between the original and the partial correlations. **Fig. 4D** shows the lowest AUC values for the action affordance model, followed by object correlations. The highest AUC difference scores are found for the GIST model, which are significantly higher than the affordance model (Mann-Whitney $U = 8.0$, $p = 0.0001$, corrected $p = 0.0015$) and the object model ($U =$

20.0, $p = 0.0010$, corrected $p = 0.0003$), suggesting a greater alignment of DNN representations with low-level features represented by the GIST model than with affordance- and object-related information in ERP responses. Overall, these results suggest that, although DNNs exhibit strong representational alignment with EEG responses, their representations correspond more closely to low-level features than to locomotive action affordances. This indicates that current DNN models may be insufficient for accurately modeling the neural processing of locomotive action affordances.

## Discussion

In this study, we examined the temporal dynamics of locomotive action affordance perception, investigating whether representations of such affordances emerge distinctly from other

scene features, and how well they align with neural responses in scene-selective regions and layer activations of DNNs.

Our results revealed that locomotive action affordances form a distinct representation, explaining ERP variability between 175 and 250 ms post-stimulus. These action affordances appear to reflect comparatively higher-level features, processed in a distinct time window after low-level GIST and object features. Scene-selective regions OPA, PPA and MPA all three explain variability in ERPs, with results indicating a potential temporal processing hierarchy: OPA processes information earlier, while PPA reaches peak explanatory power around 200 ms. Interestingly, partial correlation analyses reveals that both regions account for more ERP variability shared with affordances and objects than with GIST-based low-level features, indicating a role in processing high-level features. In contrast, while DNNs pre-trained on visual tasks also align with ERP responses, they primarily captured low-level visual feature-related variance in the ERPs, showing limited representation of locomotive action affordances.

Prior research on the temporal dynamics of navigational affordance perception has predominantly focused on synthetic (Harel et al., 2022; Djebbara et al., 2019) and exclusively indoor real-world environments (Dwivedi et al., 2024), typically defining affordances as walkable pathways. In contrast, our study adopts the paradigm of Bartnik et al. (2025), expanding the operationalization of navigational affordances to include multiple locomotive actions across both indoor and outdoor scenes. The timing of affordance perception is debated, with some studies suggesting it occurs relatively late, around 300 ms (Dwivedi et al., 2024) therefore likely building on previously extracted features. In contrast, our findings indicate that locomotive action affordances emerge earlier, around 200 ms (Harel et al., 2022; Djebbara et al., 2019). While our definition of affordances is broader than in previous studies, it remains more specific than all affordances in a scene. Nonetheless, our findings align with Greene & Hansen (2020), who reported that a much broader set of 227 scene functions peaked in neural correlation between 176 and 200 ms. In addition, the fact that our affordance representations explained unique variance in EEG responses independent of objects or GIST features suggests that affordance perception may occur in parallel, rather than using other scene features as building blocks.

Using the same stimuli as Bartnik et al. (2025) and replicating key findings, our study provided the opportunity to integrate fMRI and EEG measurements by spatiotemporal fusion (Cichy et al., 2016), offering a more complete picture of how locomotive action affordances are processed in the brain. Our results confirm that both OPA and PPA show correspondence with ERP responses to scenes, but crucially, they do so at different time points, revealing a temporal hierarchy between them. OPA, which has been linked to scene layout representation and navigational affordance processing (Bonner & Epstein, 2017; Park & Park, 2020; Epstein & Baker, 2019), exhibits an early peak around 100 ms, consistent with its role in rapid invariant encoding of scene layout (Henriksson

et al., 2019). In contrast, PPA, associated with encoding spatial layout, objects, and texture (Epstein & Kanwisher, 1998; D. B. Walther et al., 2009; Harel et al., 2013; Bastin et al., 2013), reaches peak correlations around 200 ms, overlapping with later locomotive action affordance processing.

These distinct temporal dynamics may explain why previous studies that defined navigational affordances as walkable pathways (Bonner & Epstein, 2017), relying primarily on spatial structure processed around 90–125 ms (Mononen et al., 2025), predominantly identified OPA rather than PPA. In contrast, locomotive action affordances likely engage additional scene features (e.g. textures) processed in PPA (Henriksson et al., 2019). By comparing to different representational feature spaces, we further show that these regions are not fully explained by low-level features. While GIST features are extracted early (100–150 ms) (Ramkumar et al., 2016) and significantly influence fMRI activity in PPA and OPA (Watson et al., 2014, 2017; Rice et al., 2014; Bartnik et al., 2025), our results indicate that they primarily account for early ERPs but do not overlap with affordance representations in these regions.

DNNs are widely used to model representations in visual cortex (Kietzmann et al., 2019; Storrs & Kriegeskorte, 2019), with their feature activations mapping onto various brain regions (Khaligh-Razavi & Kriegeskorte, 2014; Kriegeskorte, 2015) and effectively capturing the temporal dynamics of visual perception (Cichy et al., 2017). However, most studies have focused on object perception, leaving DNNs ability to encode navigational affordances less explored. Our results confirm that DNN representations correlate strongly with ERPs (Cichy et al., 2017), peaking around 100 ms, suggesting they effectively capture early visual processing, but do not fully capture affordance-related processing at later time points. Moreover, unlike other prior work suggesting that a DNN's training objective influences its alignment with brain representations (Dwivedi, Cichy, & Roig, 2021; Dwivedi, Bonner, et al., 2021), we find no clear benefit of models trained on e.g. scene segmentation or video, as opposed to object recognition, in explaining ERP variance to scene images. While CLIP models trained on image-text pairs exhibit one of the strongest correlations with ERP responses, the same correlation was obtained with ViT, which was trained on regular object labels, arguing against a strong benefit of language supervision on brain alignment (Wang et al., 2023; Doerig et al., 2022).

It is important to acknowledge several limitations of our study, which also highlight key directions for future research. While we extend the operationalization of affordances by using six locomotive action affordances, a key definition of affordances is that they should reflect the relations between environmental properties and the perceiver's action capabilities (Gibson, 1977; Rietveld & Kiverstein, 2014). Here more immersive tasks and naturalistic inputs (Gregorians & Spiers, 2022; Zhang & Gallant, 2020; Djebbara et al., 2021) are needed to account for the embodiment that ties a particular set of scene, object, and material properties together to an individual observer's affordances. A lack of embodiment could

also explain DNNs inability to capture locomotive affordances. Here, a way forward could be to test DNNs that are trained on human affordance labels, or to explore vision-language-action models from robotics such as OpenVLA (Kim et al., 2024), which are trained in more embodied contexts.

Finally, our locomotive action affordance annotations showed considerable correlation with object annotations, making it challenging to fully disentangle their distinct contributions to the temporal dynamics of scene perception. A more fine-grained representation of affordance-related features, such as spatial layout and textures, could help clarify how the brain processes locomotive affordances over time. In addition, our use of non-cross-validated correlation distances to construct RDMs may have introduced positive bias (A. Walther et al., 2016) in the EEG RSA results (see **Fig. S1C**); future work should replicate these results with cross-validated distance metrics. Finally, expanding the spatiotemporal fusion analysis beyond occipital, scene-selective regions to include other brain areas and examine hemisphere-specific effects could provide a more comprehensive understanding of the temporal hierarchy underlying affordance perception.

Overall, our findings demonstrate that locomotive action affordances form distinct neural representations that emerge after low-level and object features, while current DNNs fail to adequately capture these affordance-related processes.

## Data and code availability

EEG data, behavioral annotations, and analysis code are available on OSF https://osf.io/v3rcq/ and GitHub https://github.com/cgbartnik.

## Acknowledgments

This work was supported by an VENI grant (VI.Veni.194030) from the Netherlands Organisation for Scientific Research (NWO) to IIAG.

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

**Supplementary information**

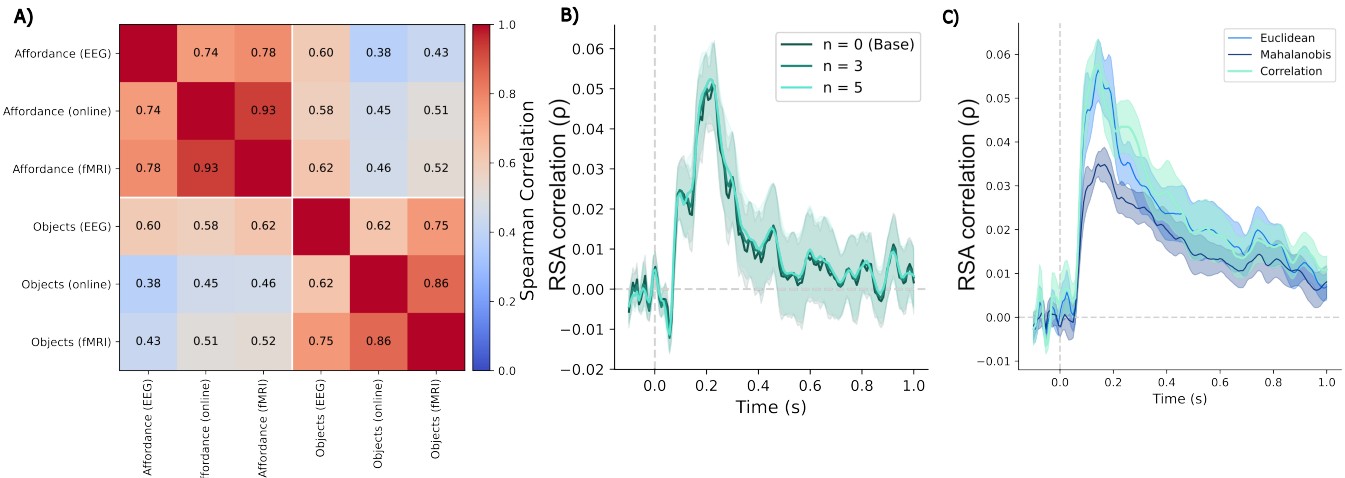

Fig. S 1: **Comparison to other behavioral datasets, effect of smoothing, and comparisons of distance metrics for computing EEG RDMs**. **(A)** Comparison (Spearman correlation) of behavioral annotations for action affordances and objects obtained during the EEG experiment to annotations done by participants in our prior online study and fMRI experiment (Bartnik et al., 2025). **(B)** Sliding window averaging of *n* timesteps to smooth the curves. Spearman correlation of the average ERP response with the action affordance RDM. **(C)** Correlation with the GIST model RDM for different distance metrics used to calculate the ERP RDMs. We decided to use the correlation distance metric.

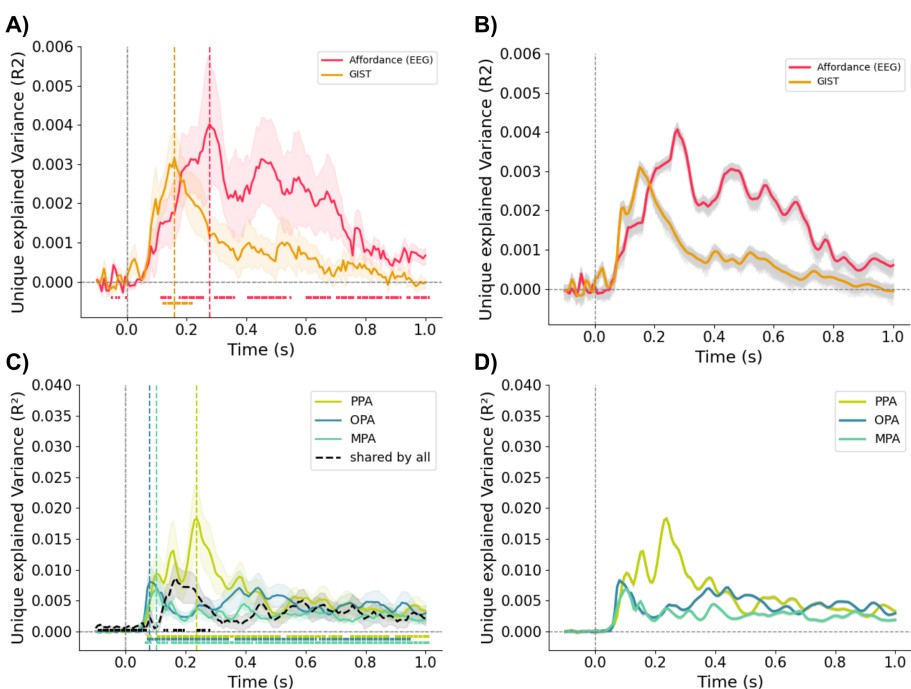

Fig. S 2: **Impact of maintaining an equal number of predictors in variance partitioning through RDM shuffling (A)** Variance partitioning of action affordance and GIST representations in the EEG signal, with model complexity controlled by maintaining an equal number of regressors through shuffling the non-target RDM. **(B)** Same analysis as in (A), but using 100 permutations. The mean is shown as a colored line; individual permutation results are shown in light gray. **(C)** Variance partitioning of scene-selective ROI RDMs (PPA, OPA, MPA) with EEG, using the same shuffling approach to control for model dimensionality. **(D)** Same as (C), but using 100 permutations.

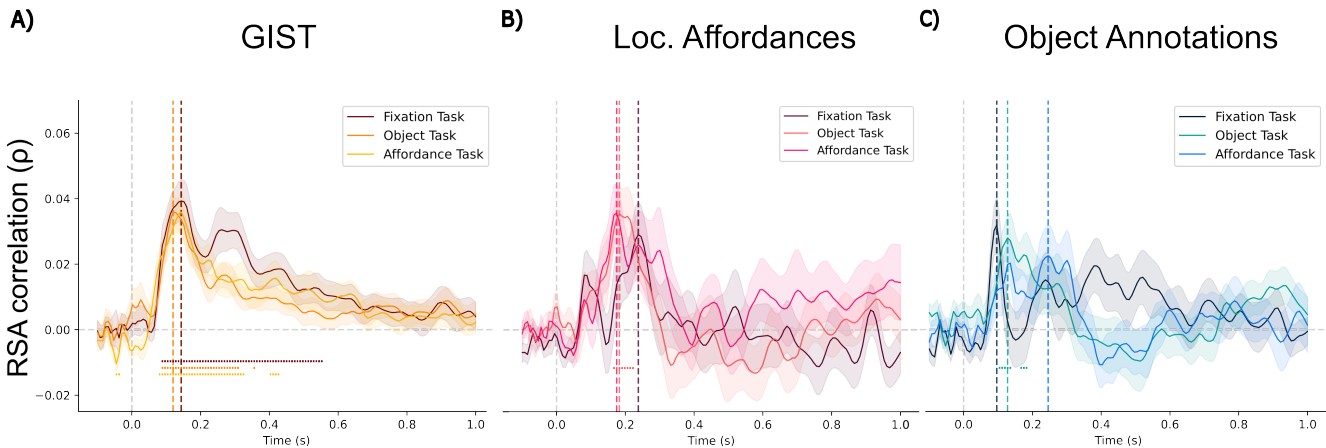

Fig. S 3: **Task-specific correlations between ERP RDMs and model RDMs. (A)** RSA correlations between RDMs, derived from ERP amplitude patterns in the 19 occipital electrodes for the three specific tasks, and GIST model features (estimated using a 256 by 256 pixel image resolution). The affordance task is shown in the lightest color, the object task in the middle color, and the fixation task in the darkest color. The lines represent the average Spearman correlation across participants, while the shaded areas indicate the standard error of the mean (SEM). Colored dots mark significant results from a one-sample t-test against zero ( p<0.05), corrected for multiple comparisons across time points using fdr correction. **(B)** Correlations for the action affordance annotations, using the same visual elements as in (A). **(C)** Correlations with the object annotations, following the same format as in (A).

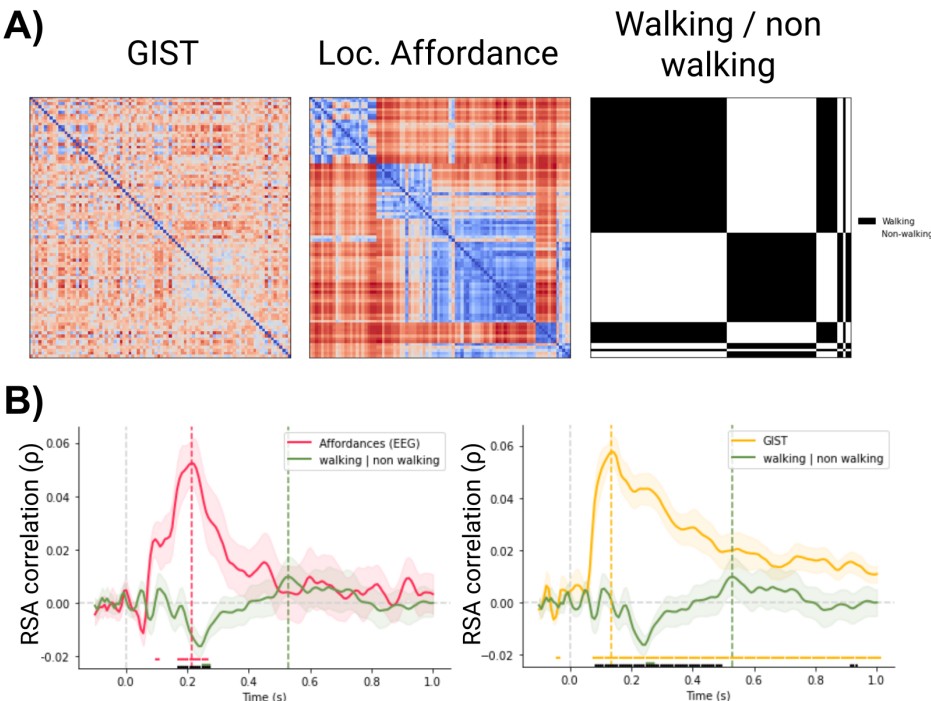

Fig. S 4: **Walking vs. Non walking (A)** Representational dissimilarity matrices (RDMs) derived from GIST features (left), locomotive action affordance ratings (middle; red = high, blue = low dissimilarity), and a binary RDM based on if walking was the highest label or another locomotive action (right; black = within-category, white = between-category). **(B)** Time-resolved, across-subject averaged Spearman correlations between ERPs and the locomotive affordance RDM next to the walking/non-walking RDM (left), and between ERPs and the GIST feature RDM alongside the walking/non-walking RDM (right). Shaded areas represent the SEM across participants; significant time points (FDR-corrected) are indicated by dots, and vertical dashed lines mark peak correlation times.

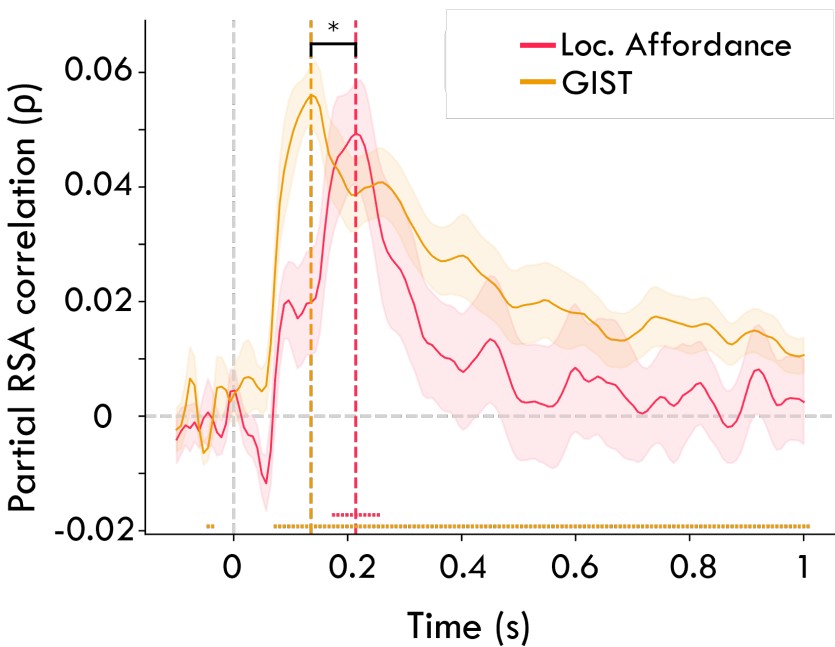

Fig. S 5: **Partial correlation on GIST and ERP correlations**. Partial correlations between GIST and ERP RDMs, controlling for locomotive action affordance representations, and vice versa.; shaded areas indicate SEM. Significant time points (fdr-corrected) are marked with dots, and vertical dashed lines highlight peak correlation times.

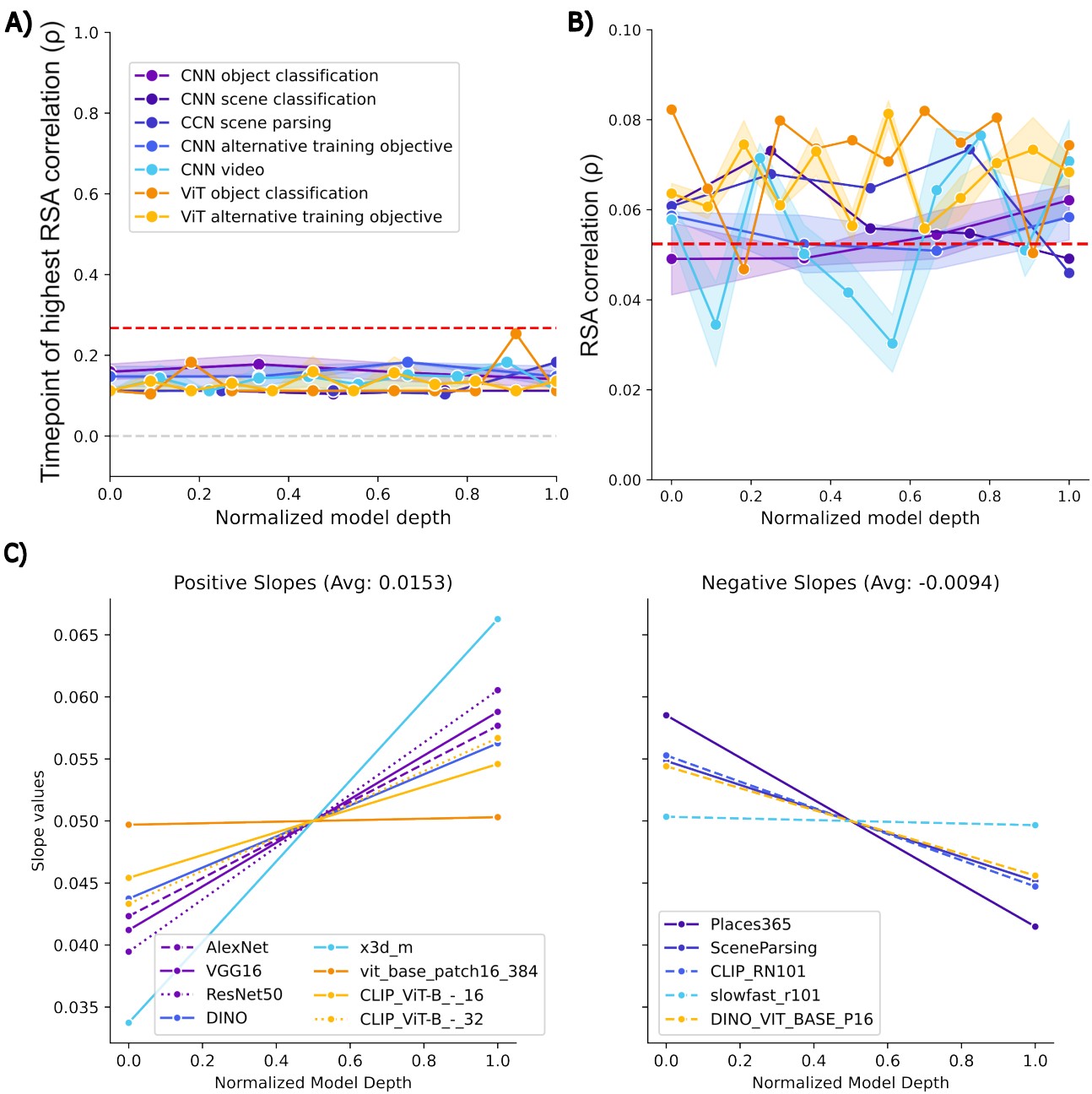

Fig. S 6: **Impact of DNN model depth on correlations with the ERPs.** **(A)** Highest correlating time points by normalized model depth for each group of model. The red dashed line indicates the highest average timepoint for the affordance space. Lines indicate the mean correlation for each model group with the shaded area indicating SEM of the mean for each group if multiple models were contained. If models in each group had different numbers of layers we reduced the numbers to the lowest common number of layers. The grey dashed line at zero indicated the image onset. **(B)** Highest Spearman correlation by normalized model depth and model group. Same elements as in (A). **(C)** Regression lines for the Spearman correlations for each DNN model. In the first Panel we show the positive slopes and in the second panel we show the negative slopes.

