# OpenReview forum: "Temporal misalignment in scene perception: Divergent representations of locomotive action affordances in human brain responses and DNNs"
_ccneuro.org/CCN/2025/Proceedings — CCN 2025 Proceedings asProceedingsPoster_

### Official Review · Reviewer_JUxZ · 2025-03-30
**Interesting and well-made paper with room for improvement in layout and description of methods**

**Soundness:** 3
**Clarity:** 2

**Comments:**

The reviewed paper investigates the neural representation of locomotive action affordances in response to diverse visual scenes mainly via analysis of EEG-derived ERP data and corresponding behavioral affordance- and object annotations through Representational Similarity Analysis.  Locomotive action affordance representations were thereby found to explain ERP variability within 200 ms of visual processing, while contributions to EEG responses from objects and low-level stimulus properties occurred earlier in time. Through spatiotemporal fusing with data from an earlier fMRI study, it was further found that the medial-, parahippocampal-, and occipital place areas were involved in locomotive action affordance representation. Additional analyses showed that current deep neural networks are limited in their ability to model affordance perception.

To set the rest of the review into context, I am only superficially familiar with EEG data analysis, so I cannot comment on the viability of the methods used here for recording and analysis in relation to ERPs.

In my opinion, this is an interesting paper, and I think it fits the scope of the conference well. The authors make use of Representational Similarity Analysis as a tool to compare results over different modalities, and I think it was used well in this case. All analyses follow an internal consistency and in the discussion part, the authors place their results in the context of previous research findings and draw meaningful conclusions regarding potential knowledge gain from their study.

I do have two main points of criticism / suggestions for improvement:

1. I would appreciate a more in-depth explanation on which parameter(s) the distance was calculated between pairs of behavioral responses, both for affordances and objects. As of now, I am not sure how e.g. an action affordance was calculated to be more or less dissimilar to one of the other five affordances, except for the method of calculation (Euclidean distance).

2. For this kind of paper, the layout might not be ideal. Having the methods and material at the end is okay but made it harder for me to grasp the different analyses and their interplay.  This is also the case for the appendix. For example, SEM is first introduced as an abbreviation in figure caption 2A but explained in supplementary figure 1A. I do acknowledge that this was in parts owed by the 8-page limit.
As another minor point, if I understood the formatting instructions correctly, citing for cases of naming the author in text needs a revision, with only the year of publication in parenthesis (examples in lines 392 & 399).

Nevertheless, I think it is a good paper, and I would judge it to be not only of interest for the cognitive- and neuroscience community. Due to its implication of affordance- and object-specific (temporal) processing differences, results could also be of interest for the computer vision community. The authors of the paper demonstrate this by their analysis of correlation with results from DNNs.

**Expertise:**

1

**Interest:**

3

---

> ### Author Rebuttal · Authors · 2025-04-14
>
> Thank you for your constructive review. We appreciate your positive assessment and helpful feedback. Below, we address each of your points. All changes in the revised manuscript are marked in blue.
>
> ---
>
> **Clarification on how behavioral distances were computed for affordances and objects**
>
> We thank the reviewer for highlighting the need for a more detailed explanation of how behavioral dissimilarities were calculated. To clarify: for each of the behavioral annotation tasks (i.e., six locomotive action affordances and six object categories), participants were asked to indicate which labels applied to a given scene.
> For each scene, we computed the proportion of participants who endorsed each label, resulting in a behavioral response vector (e.g., a 6-dimensional vector for affordances, where each dimension reflects the proportion of participants selecting a given affordance for that scene). These vectors formed the basis for calculating dissimilarities.
> Specifically, representational dissimilarity matrices (RDMs) were constructed by computing the Euclidean distance between these label-proportion vectors across all pairs of scenes, separately for each task. This yielded five 90 × 90 symmetric RDMs—one per annotation task—where each matrix entry reflects how dissimilar two scenes were in terms of their perceived affordances or objects, based on participant responses. We have revised the *Behavioral Annotations* paragraph in the Methods section (line 589).
>
> ---
>
> **Abbreviation and citations:**
>
> We agree that it can be challenging to follow the results without prior familiarity with the methods. However, this ordering—placing Methods and Materials at the end—is a common convention in our field, and we chose to follow it for consistency. That said, we now introduce SEM as “standard error of the mean (SEM) across participants” in Figure 2 of the main text, and we have corrected the citation formatting as noted.

---

> > ### Comment · Reviewer_JUxZ · 2025-04-22
> >
> > I appreciate the clarifications made by the authors.

---

### Official Review · Reviewer_p54C · 2025-03-31
**The study thoroughly investigates the emergence of action affordances in the brain, revealing a significant gap in current vision DNNs.**

**Soundness:** 3
**Clarity:** 3

**Comments:**

The present study asks whether the perception of action affordances—identifying actions to navigate an environment—is a late neural process that builds upon earlier processing of low-level visual features, object information, and semantics. Using spatiotemporal fusion of EEG and fMRI responses to images of scenes, the authors show that representations of action affordances emerge in posterior regions within 200 ms only after processing of low-level features and objects. Specifically, action affordances first appear in scene-selective Occipital Place Area (OPA), followed by Parahippocampal Place Area (PPA). Interestingly, representations from vision DNNs show little overlap with affordance-specific representations in EEG responses, possibly due to their lack of embodiment. This suggests that representations of action
affordances emerge earlier in the brain than previously expected, indicating a parallel process to other feature-building processes.

The study convincingly highlights a shortcoming of state-of-the-art vision DNNs in capturing a crucial aspect of human visual perception—action affordances. This insight potentially paves the way for developing more human-like DNNs, although the study does not offer a solution at this point. The authors carefully distinguish the roles of low-level and object processing in the brain from those of action affordances, shedding new light on the computational underpinnings of this subfield of visual neuroscience. The manuscript is well-structured, with clear rationale and methods that are easy to follow. The analyses are sound and comprehensive, and the figures are informative. I have only a few suggestions for improving the methods.

Major comments
1. In the variance partitioning approach, it appears that one RDM at a time was removed from the regression model. This might lead to a drop in variance explained due to the smaller number of regressors included, rather than the importance of the regressor itself. Could the authors repeat the procedure but instead shuffle or time-shift regressors of interest?

Minor comments
1. “Low-level features, such as GIST, are extracted between 90–150 ms […]” (lines 56 to 57) Please provide examples of these low-level features—does GIST represent features such as edges?
2. Are RSAs cross-validated to ensure the generalizability of results?

**Expertise:**

2

**Interest:**

3

---

> ### Author Rebuttal · Authors · 2025-04-14
>
> We thank the reviewer for their constructive and insightful feedback. We appreciate the time and effort taken to evaluate our work and have addressed each of the points below. All changes are marked in blue in the revised manuscript.
>
> ---
> **Variance partitioning:**
>
> To address this concern, we modified our analysis to maintain the same number of regressors in the model while selectively removing the information content from the regressor of interest. Specifically, when computing the unique variance of a single RDM, we added a shuffled version of the second RDM—thereby preserving the total number of regressors while disrupting the meaningful structure in the second RDM. This ensures that the variance captured reflects only the contribution of the regressor of interest.
> For both the 2 regressor and the 3 regressor variance partitioning approaches we include additional shuffled regressors, thereby preserving the total number of regressors. As shown, in the Supplementary Figure 5 the curves remain highly similar in both shape and magnitude, indicating that the drop in unique variance is indeed due to the removal of meaningful information, rather than a reduction in model dimensionality.
>
> To further support this, we performed 100 permutations with different shuffle seeds and averaged the resulting curves. The spread across permutations confirms that the effect of regressor count is minimal, and the impact of removing meaningful variance remains consistent with our original findings.
>
> This approach aligns with methods used in recent literature (e.g., Dwivedi et al., 2024, https://doi.org/10.1038/s41598-024-55652-y), reinforcing the validity of our original procedure.
>
> ---
>
> **GIST Clarification:**
>
> This point was also raised by another reviewer, and we have now expanded the description of the GIST model accordingly. To provide a clearer intuition of the features it captures, we revised the Results section starting from line 180.
>
> ---
>
> **Cross-validation in RSA:**
>
> The RSA results were not cross-validated, consistent with prior work (e.g., Bonner & Epstein, 2017, https://doi.org/10.1073/pnas.1618228114; Dwivedi et al., 2024, https://doi.org/10.1038/s41598-024-55652-y). However, p-values were FDR-corrected at an alpha level of 0.05.

---

> > ### Comment · Reviewer_p54C · 2025-04-17
> >
> > I appreciate the authors' thorough clarifications and revisions to the manuscript.
> >
> > The variance partitioning analysis has been improved by controlling the number of regressors, and the results remain robust. Consequently, I have increased the score for soundness.
> >
> > Regarding RSA, it is generally recommended to cross-validate results (Walther et al., 2016) as noise in the data can introduce a positive bias to distances.

---

> > > ### Author Response · Authors · 2025-04-20
> > >
> > > Thank you for increasing the score for soundness based on the revisions we made, and for your further feedback.
> > >
> > > We agree it is good practice to cross-validate results by using e.g. the cross-validated Mahalanobis distance proposed in Walther et al. (2016) for constructing the RDMs, rather than average correlation distance across all measurements. In fact this work prompted us (see line 586) to report a version of the RSA analysis that uses this distance metric in Fig S2, where we compare it to Euclidean and correlation distance. While the shape of the correlation time course as well as its peak are similar across metrics, it is lower for the Mahalanobis distance, possibly indicating some contribution of positive bias to our results.
> > >
> > > To acknowledge this, for the camera-ready version of the manuscript, we will point the reader to this result once more and add a sentence to the second paragraph of the limitations section of the Discussion (starting on lines 478) as follows:
> > >
> > > Finally, our locomotive action affordance annotations showed considerable correlation with object annotations, making it challenging to fully disentangle their distinct contributions to the temporal dynamics of scene perception. More fine-grained representation of affordance-related features, such as spatial layout and textures, could help clarify how the brain processes locomotive affordances over time. **In addition, our use of non-cross-validated correlation distances to construct RDMs could have introduced positive bias (Walther et al., 2016) in the EEG correlations (see Fig S2C); future work should replicate these results with cross-validated distance metrics.** Furthermore, expanding the spatiotemporal fusion analysis beyond occipital, scene-selective regions to include other brain areas and examine hemisphere-specific effects could provide a more comprehensive understanding of the temporal hierarchy underlying affordance perception.

---

### Official Review · Reviewer_stap · 2025-03-31
**Temporal misalignment in scene perception: Divergent representations of locomotive action affordances in human brain responses and DNNs**

**Soundness:** 3
**Clarity:** 2

**Comments:**

This is a good, and reasonably rigorous study looking at the temporal emergence of action affordances, based on subjects viewing various scenes which lead to different actions (walk, swim, bike etc\). Behavioural annotations are mapped to EEG responses using an RSA approach and used to compare signals from different brain regions associated with scene perception. OPA appears to process affordances earlier than PPA, and this is combined with fMRI ROI analysis based on a previous data set. The responses are dissociable from object specific features, of features derived from deep NNs. Overall, the results help establish a spatiotemporal pathway for affordance perception.

Comments: the GIST model is not fully explained and confused me slightly.
I wonder to what extent the results are driven by walking vs non-walking affordances.
How is laterality accomodated,

**Expertise:**

2

**Interest:**

2

---

> ### Author Rebuttal · Authors · 2025-04-14
>
> Thank you for your thoughtful and positive review. We appreciate your constructive feedback and have addressed your comments in the revised manuscript. All changes are marked in blue.
>
> ---
>
> **GIST model:**
>
> We agree that the explanation of the GIST model was insufficient, especially given that it is a well-known concept in the field. Since it plays a central role in our work and is important for broader understanding, we have revised parts of the Results section (starting from line 180) to provide a clearer intuition of the types of features captured by the GIST model.
>
> ---
>
> **Walking vs. non-walking:**
>
> Your question relates well to prior work focused on walking-related affordances (e.g., Bonner & Epstein, 2017, https://doi.org/10.1073/pnas.1618228114; Harel et al., 2022, 10.1162/jocn_a_01810; Dwivedi et al., 2024, https://doi.org/10.1038/s41598-024-55652-y). Our study expanded this scope to include other action modalities (e.g., swimming, biking), though we did not explicitly assess the impact of this broader range.
>
> To address this, we created a binary RDM contrasting walking vs. non-walking and computed RSA correlations with the EEG signal. We added Supplementary Figure 3, showing overall lower correlations and for some time points negative ones.
> Interestingly, we still observe a significant peak around 250 ms post-stimulus onset, suggesting that affordance-related distinctions along the walking dimension are reflected in the neural signal at this time point. While negative correlations in RSA are challenging to interpret, they may indicate that scenes similar in the binary model (e.g., all walking) evoke distinct EEG patterns—implying meaningful variation in neural representations within that affordance category.
>
> Importantly, these new results support the notion that the temporal dynamics we observe are not solely driven by the walking/non-walking distinction, but rather reflect a richer, multidimensional structure of action affordances encoded in neural activity.
>
> ---
>
> **Laterality:**
>
> In this study, we combined EEG signals across occipital electrodes and did not examine hemispheric differences. However, we agree that laterality is a relevant aspect. We now mention in the Discussion (line 487) that future work could explore hemisphere-specific effects and lateralized contributions of scene-selective regions. We added possible future directions to our discussion.

---

### Official Review · Reviewer_BgGr · 2025-04-01
**Excellent manuscript.**

**Soundness:** 3
**Clarity:** 3

**Comments:**

This study is of exceptional quality, bridging cutting edge approaches in cognitive neuroscience and compuational neuroscience and connecting to some topics in AI research.
I do not have many suggestions to improve this manuscript.
The introduction clearly situates this work in the field and presents the rationale of the study.
The methods are clearly described and the results section and figures perfectly presents the findings.
The discussion is also very well written and organized.
* It would be interesting to discuss the "Deep Global Workspace Theory" developped by Vanrullen and colleagues (see this review: https://doi.org/10.1016/j.tins.2021.04.005, but also more recent work by the team of R. Vanrullen) as this frameworks proposes to merge multiple modalities and motor learning into a single model. Could this overcome the limitations in affordance representation that the author discuss?

**Expertise:**

3

**Interest:**

3

---

> ### Author Rebuttal · Authors · 2025-04-14
>
> Thank you for your enthusiastic and positive review, and for the thoughtful suggestion to consider the relevance of the Deep Global Workspace (DGW) theory in addressing the limitations in affordance representation that we raised.
>
> ---
>
> The DGW framework builds on the Global Workspace Theory (GWT), extending it by integrating multiple specialized DNN modules through a shared latent space—the “workspace”—to support multimodal and goal-directed processing. GWT has been extensively studied in the context of EEG, where the concept of “ignition” refers to widespread cortical activation, particularly involving frontal regions, associated with conscious access. This process is typically linked to the P3 component around 300 ms post-stimulus (for more details see e.g. Mashour et al., 2020, https://doi.org/10.1016/j.neuron.2020.01.026).
>
> In contrast, our findings highlight affordance-related activity in an earlier time window (175–250 ms), likely preceding such global integration. We therefore interpret our results as reflecting activity within specialized modules—such as a vision module—prior to their integration into a global workspace. This interpretation is supported by the absence of task effects in our data, suggesting that the observed affordance processing occurs automatically and independently of task demands, similar to prior research we mention in the Discussion. Additionally, our analyses were restricted to occipital electrodes, limiting our ability to draw conclusions about broader cortical involvement. Nonetheless, the DGW framework presents a promising direction for future research on affordance perception, particularly in more embodied, multimodal, and goal-driven contexts. We chose not to expand on this in the Discussion, as we feel it goes beyond the scope of the current study and we are constrained by the 8-page limit.

---

> > ### Comment · Reviewer_BgGr · 2025-04-22
> >
> > Thank you for the thoughtful reply to my suggestion. I agree that temporal extent of the effects studied in this manuscript do not overlap with the effects classically studied within the GWT framework.

---

### Meta-Review · Area_Chair_aasj · 2025-05-04

**Ccn Recommendation:** Accept as Proceedings

**Metareview:**

This submission received consistently positive evaluations from all four reviewers, who highlighted its strong interdisciplinary interest, methodological rigor, and overall clarity. The reviewers raised thoughtful and constructive comments, and the authors engaged thoroughly and professionally with each point.

All major reviewer concerns were addressed convincingly. Reviewer p54C’s question about potential confounds in the variance partitioning analysis was met with a reanalysis using shuffled regressors, demonstrating that model dimensionality was not driving the results. Similarly, the suggestion to use cross-validated RSA methods (Walther et al., 2016) was acknowledged and incorporated via Mahalanobis distance analyses, with results reported and discussed appropriately. Concerns about potential walking-specific effects and hemispheric laterality (raised by reviewer stap) were addressed with an additional binary RDM analysis and discussion of future directions. The authors also clarified the computation of behavioral dissimilarity measures in response to reviewer JUxZ’s request.

On clarity, while some reviewers initially noted that certain methodological descriptions (e.g., the GIST model, abbreviation use) were underdeveloped, these were effectively revised. Reviewers expressed satisfaction with the authors’ updates, and no remaining concerns were noted post-rebuttal. One reviewer (p54C) explicitly upgraded their soundness score following the revisions.

In summary, the authors were responsive and thorough in their revisions, and the reviewers’ points were all substantively addressed. The paper presents a well-executed and timely contribution at the intersection of neuroscience and AI, with a strong foundation in both empirical data and theoretical framing. Based on the strength of the work and the constructive, resolved review process, I recommend acceptance.

**Summary:**

All reviewers considered the paper highly interesting, especially for interdisciplinary audiences across cognitive neuroscience, computational neuroscience, and AI. Reviewer JUxZ additionally emphasized its relevance for computer vision due to insights into affordance representation and limitations of current DNNs.

The study’s soundness was praised for its integration of EEG, fMRI, behavioral annotation, and RSA, which together supported robust conclusions. Reviewers BgGr and stap highlighted the theoretical significance of the observed temporal dissociation. Reviewer p54C raised concerns about whether variance partitioning results could be affected by model dimensionality, and suggested cross-validation in RSA. Reviewer stap asked about walking-specific affordances and hemispheric effects, while JUxZ requested clearer explanation of behavioral dissimilarity computations.

The authors addressed these by rerunning variance partitioning with shuffled regressors (added in Supplementary Fig. 5), applying cross-validated Mahalanobis distance to RSA (Fig. S2), analyzing a binary walking/non-walking RDM, and clarifying behavioral dissimilarity methods in the text. Reviewer p54C increased their Soundness score based on these revisions.

In terms of clarity, two reviewers rated the manuscript exceptional for its accessibility and structure, though some noted initial issues with the GIST model explanation and undefined abbreviations like SEM. These were all resolved, and no clarity concerns remained. One reviewer’s score was explicitly raised, and no major issues persisted after revisions.

**Expertise:**

3